# Strength, extent and duration of secondary hyperalgesia induced by high-frequency electrical stimulation of the foot compared to the volar forearm of healthy human volunteers

Louisien Lebrun[1]*, Cédric Lenoir[1], Caterina Leone[2], Emanuel N. van den Broeke[1], Ombretta Caspani[3], Andreas Schilder[4], Bernhard Pelz[5], Andrea Truini[2], Rolf-Detlef Treede[3,6], André Mouraux[1]

1 Institute of Neuroscience (IONS), Université catholique de Louvain (UCLouvain), Brussels, Belgium, 2 Department of Human Neurosciences, Sapienza University, Rome, Italy, 3 Mannheim Center for Translational Neurosciences, Medical Faculty Mannheim, Heidelberg University, Mannheim, Germany, 4 Department of Orthopaedic and Trauma Surgery, Medical Faculty Mannheim, Heidelberg University, Mannheim, Germany, 5 MRC Systems GmbH, Heidelberg, Germany, 6 Department of Psychiatry and Psychotherapy, Central Institute for Mental Health, Medical Faculty Mannheim, Heidelberg University, Mannheim, Germany

* louisien.lebrun@student.uclouvain.be

## Abstract

High-frequency electrical stimulation (HFS) of the skin using a multi-pin electrode activating epidermal nociceptors is used to explore spinal central sensitization in humans. Most previous studies applied HFS to the volar forearm. To prepare for clinical applications in which HFS could be applied to different body sites, this study compared the secondary hyperalgesia induced by stimulation of the foot dorsum vs. the forearm in 32 healthy volunteers. HFS consisted in five 1-s trains of 100 Hz pulses (inter-train interval: 10 s; intensity: 20x detection threshold) delivered via a novel electrode optimized for stimulation of different body sites (ten 0.25 mm pins in a 5-mm circle). Pinprick sensitivity was assessed before HFS and 30–240 minutes after HFS, at the treated site and the corresponding contralateral site. The area of hyperalgesia was quantified. HFS to the foot induced a significant increase in pinprick sensitivity of the surrounding skin, similar in magnitude to the increase at the forearm, and decaying similarly over time (half-lives 150 vs. 221 min). The radius of secondary hyperalgesia was smaller at the foot (22 mm) compared to the forearm (38 mm, $p < 0.001$), and decreased more rapidly over time (53 vs. 87 min, $p < 0.01$). Our results show that strength of HFS-induced secondary hyperalgesia can be used as indicator of spinal central sensitization across body sites, and thereby profile patients with localized or regional pain conditions. The size of the area of hyperalgesia may depend on innervation density and peripheral receptive field sizes.

**Data availability statement:** The data underlying the results presented in the study are available from https://osf.io/v5tbz/?view_only=42a145ccb7334a119b833ea77df47612.

**Funding:** This study was supported by a funding from the Innovative Medicines Initiative 2 Joint Undertaking under grant agreement No [777500]. This Joint Undertaking receives support from the European Union's Horizon 2020 research and innovation program and EFPIA. www.imi.europa.eu; www.imi-paincare.eu. The statements and opinions presented here reflect the author's view and neither IMI nor the European Union, EFPIA, or any Associated Partners are responsible for any use that may be made of the information contained therein. The funders had no role in study design, data collection and analysis, decision to publish, or preparation of the manuscript.

**Competing interests:** The authors have declared that no competing interests exist.

## Introduction

An important property of the nociceptive system is its propensity to sensitize when exposed to repeated nociceptive input or following tissue injury [1–3]. Sensitization of the nociceptive system can involve a combination of changes at the level of peripheral nociceptors leading to greater sensitivity to noxious stimuli, and changes at several levels of the central nervous system [4] leading to an enhanced synaptic transmission of nociceptive input that has been described extensively at the level of the dorsal horn in animal studies (spinal central sensitization) [5,6]. A perceptual consequence of sensitization is hyperalgesia or increased sensitivity to painful stimuli [7]. Following skin injury, hyperalgesia develops both within the injured area (primary hyperalgesia) and in the surrounding uninjured skin (secondary hyperalgesia) [8,9]. While both peripheral and central sensitization may contribute to the development of primary hyperalgesia, secondary hyperalgesia is thought to result exclusively from central sensitization [9,10]. A hallmark feature of secondary hyperalgesia is a prominent increase in sensitivity to mechano-nociceptive input, which can be characterized by evaluating the sensitivity to mechanical pinprick stimulation of the skin [11].

Several methods have been described and validated as means to induce central sensitization experimentally in human volunteers [12,13]. Among these methods, high-frequency electrical stimulation (HFS) of skin nociceptors using a multi-pin electrode has gained more prominence in recent years [3,14,15]. This is due to several advantages over other methods such as topical application or intradermal injection of capsaicin. Inducing central sensitization using HFS (i) is not dependent on the operator and (ii) generates a robust increase in mechanical pinprick sensitivity lasting several hours, without (iii) any confounding spontaneous on-going sensation during the post-induction period. Furthermore, it is (iv) a very brief procedure (v) that does not require administration of any substance such as capsaicin.

Most previous studies have characterized HFS-induced secondary hyperalgesia at the level of the volar forearm [3,14,15]. The area of HFS-induced hyperalgesia was reported to be four times smaller at the distal hand dorsum compared to the proximal forearm (10 vs. 39 cm²) [16]. Two studies that characterized HFS applied to the foot reported conflicting evidence on the magnitude of the induced secondary hyperalgesia: the increase in pinprick pain was not significant in one study [17] and significant in the other [18].

The present study aimed to assess whether the magnitude or area of HFS-induced secondary mechanical hyperalgesia are suitable indicators of the susceptibility to develop (spinal) central sensitization in humans. As a step towards clinical use of this surrogate model of long-term potentiation (LTP), we revisited HFS-induced hyperalgesia on the foot dorsum, because this area is also testable by the nociceptive RIII reflex [17,18], which would allow comparisons with rodent data.

Furthermore, future studies could exploit this approach to explore the dependency of central sensitization on peripheral and central factors differentiating body sites such as differences in skin innervation density, receptive field sizes and cortical representation. Moreover, the ability to investigate variations in the susceptibility to develop central sensitization across body sites would be of interest to explore the pathophysiology of localized or regional chronic pain conditions.

## Methods

### Experimental design

32 healthy volunteers (15 women and 17 men, aged between 20 and 30 years; mean 23.7 years) were included between 11th of April 2019 and 2nd of March 2022. The study was approved by

the local ethics committee (comité d'éthique hospitalo-facultaire des Cliniques universitaires Saint-Luc-UCLouvain; B403201316436). All participants provided written informed consent and received a financial compensation. The experiment was conducted according to the Declaration of Helsinki. The study protocol was not published in advance.

Half of the participants (N = 16; 8 women and 8 men, aged 20-30 years; mean 23.9 years) were randomly assigned to receive HFS on the volar forearm and the other half (N = 16; 7 women and 9 men, aged 20-28 years; mean 23.5 years) received HFS on the lateral side of the foot dorsum. For both groups, the arm/foot onto which HFS was applied was balanced according to their handedness, assessed using the Flinders Handedness survey [19]. The chosen sample size (N = 16 per group) was not based on a sample size calculation as effect sizes were not known for this exploratory study. Our sample size was similar to the size of the samples used in several previous studies assessing differences in HFS-induced secondary mechanical hyperalgesia such as differences related to the frequency of stimulation (15 and 16 participants in van den Broeke et al., J Neurophysiol 2019 [15]) or differences related to the pattern of HFS stimulation (15 participants in Gousset et al., J Neurophysiol 2020 [20]).

The inclusion criterion was being aged 18–40 years. Exclusion criteria were suffering from cardiac or neurological disorders and practicing sport requiring intense use of forearms or foot dorsum (e.g., volleyball, handball, football, soccer). Moreover, participants with a history of traumatic injury of the upper limb, lower limb or head, participants with a dermatological condition involving the forearm or foot, and participants having participated in a previous experiment involving HFS were excluded. Finally, participants were asked to have slept at least 6 hours the night before the experiment, and to refrain from recreational drugs and medication including analgesics for a minimum of three days preceding the experiment, except for oral contraception.

## Induction of secondary hyperalgesia using HFS

For stimulation of the volar forearm, participants were seated on a comfortable chair with their arms resting on a table, volar forearms facing upwards. The HFS electrode was placed 10 cm distal from the cubital fossa (dermatomes C6-C7-C8-T1) [21]. For stimulation of the foot, they laid on a comfortable examination bed, and the HFS electrode was positioned on the dorsolateral side of the foot, 3 cm distally from the lateral malleolus and 2 cm proximally from the sole of the foot (dermatomes L5-S1) [21].

HFS was applied using a recently developed multi-pin electrode (EPS-P10, MRC Systems GmbH, Heidelberg) (see Fig 1). The electrode consists of a multi-pin cathode designed to preferentially activate cutaneous free nerve endings (10 blunt tungsten pins arranged on a circle with a diameter of 5 mm) and a flat anode (24 × 20 mm conductive gel pad) linked to the cathode using a flexible connector. Each pin has a diameter of 0.25 mm and protrudes by 0.65 mm over the base of the electrode. The anode and cathode are linked by a flexible connector. The stimulation consisted of five trains of 100 Hz charge-compensated biphasic pulses (2 ms pulse followed by a 4 ms compensation pulse of opposite polarity having half the intensity of the first pulse), delivered using a constant-current electrical stimulator (Digitimer DS5; Digitimer, Welwyn Garden City, United Kingdom) controlled by a National Instruments digital-analogue interface (NI6343, National Instruments, Austin, Texas, USA) [15]. Each train lasted 1 second (100 pulses/train). The time interval between each train onset was 10 s. The intensity of stimulation was set to 20 times the detection threshold to a single pulse, which was estimated in each participant at the beginning of the experiment, using the method of limits (forearm: 0.12 ± 0.06 mA; foot: 0.30 ± 0.10 mA [mean ± SD]).

## Assessment of the HFS-induced change in mechanical pinprick sensitivity

The sensitivity to mechanical pinprick stimuli was assessed 7 times (before HFS, and 30, 60, 90, 120, 180 and 240 minutes after HFS) at the left and right volar forearms or the left and right foot dorsum using a custom 0.25 mm flat-tip pinprick probe exerting a 128 mN force [15] (Fig 1). Participants were asked to close their eyes during the pinprick sensitivity measures. At each time point, three pinprick stimuli were applied perpendicularly to the skin within a 2 cm circle surrounding the area of HFS stimulation at the sensitized limb, and on the same location of the non-sensitized contralateral limb. After each stimulus, participants were requested to report the intensity of the pinprick-evoked sensation using a numerical rating scale ranging from 0 (no perception) to 100 (maximal imaginable pain), with 50 marking the limit between non-painful and painful sensations. This scale, which allows rating the intensity of both painful and non-painful stimuli, has been used in several previous experiments because the pinprick stimuli are often not perceived as painful at baseline [22–25]. To avoid sensitization of the skin by repeated mechanical pinprick stimulation, the probe was not applied twice on the same location. Stimuli were applied every 5 seconds. The order of the testing (HFS-treated limb or contralateral limb) was balanced across participants and remained identical across the different time points. Pinprick sensitivity at each time point and limb was expressed as the mean of the three ratings.

## Assessment of the area of HFS-induced secondary hyperalgesia

The same 128 mN pinprick probe was used to map, at each post-HFS timepoint, the area of increased mechanical pinprick sensitivity surrounding the site of HFS at the HFS-treated limb. The stimuli were applied along 8 directions separated by a 45° angle. Participants were asked to close their eyes during the mapping procedure. Starting far from the HFS-treated skin and moving towards the center, stimuli were applied every 5 mm. Participants were asked to report when the sensation increased and turned to be more pricking, and the corresponding stimulation location was marked with a felt-tip pen [23,25–27]. For each of the 8 directions, we then measured the distance between the center of the HFS electrode location and the mark. The area of secondary hyperalgesia was computed as the area of a polynomial 2D closed curve fitted onto the 8 marked locations ('interpclosed' function version 3.0 using cubic spline interpolation; Santiago Benito 2021, Matlab Central File Exchange). The radius of the area of increased pinprick sensitivity was then computed as the square root of the area divided by the square root of pi.

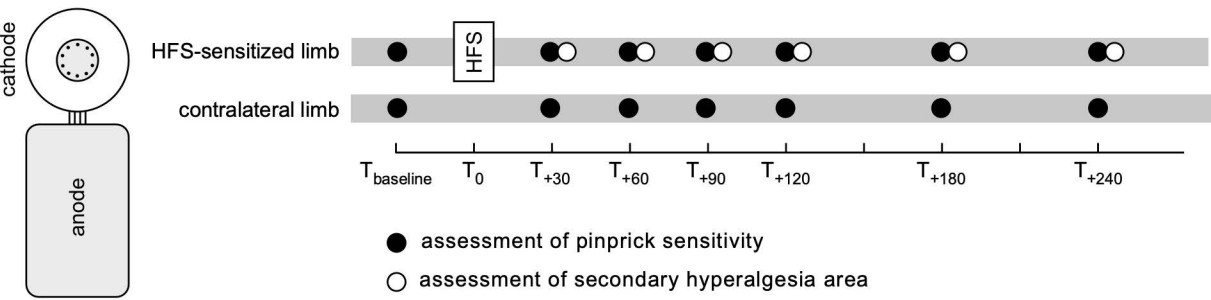

**Fig 1. High-frequency electrical stimulation (HFS) of the skin was delivered using a novel electrode designed to preferentially activate cutaneous nociceptors, consisting of a multi-pin cathode (10 blunt tungsten pins arranged in a 5 mm diameter circle) and a large-surface anode (24 × 20 mm) bound together using a flexible connector.** Participants were exposed to HFS delivered either to the volar forearm or the foot dorsum. Pinprick sensitivity was assessed at one time point before HFS ($T_{baseline}$) and six time points ranging from 30 minutes to 4 hours after HFS ($T_{+30}$, $T_{+60}$, $T_{+90}$, $T_{+120}$, $T_{+180}$, $T_{+240}$), at the HFS-sensitized limb and the contralateral non-sensitized limb. The extent of the HFS-induced area of secondary hyperalgesia area was assessed at each timepoint after HFS application, at the HFS-sensitized limb.

## Statistical analyses

Statistical analyses were performed using JASP (Version 0.14.1.0 for Windows). Threshold for significance was set at 0.05. Results are reported using median and interquartile range (IQR).

The primary aim of our study was to evaluate and compare the strength, duration and spatial extent of HFS-induced secondary hyperalgesia induced at the forearm and at the foot.

**HFS-induced increase in pinprick sensitivity at the forearm or at the foot.** To determine whether HFS delivered to the forearm induced a significant increase in pinprick sensitivity at the sensitized forearm, and whether HFS delivered to the foot induced a significant increase in pinprick sensitivity at the sensitized foot, the change in pinprick ratings 30 minutes after HFS at the HFS-treated limb was compared to the change in pinprick ratings 30 minutes after HFS at the contralateral limb ([HFS limb: $T_{+30}$ minus $T_{baseline}$] vs. [contralateral limb: $T_{+30}$ minus $T_{baseline}$]), as a plateau is typically reached 30 minutes after HFS [28]. The Shapiro-Wilk test indicated a deviation from normality for those change scores both at forearm and at the foot. Therefore, both within-subject comparisons were conducted using a paired Wilcoxon test.

**Comparison of the HFS-induced increase in pinprick sensitivity at the forearm and foot.** For both limbs, we first expressed the HFS-induced increase in pinprick sensitivity as the change in rating ($T_{+30}$ minus $T_{baseline}$) at the HFS-sensitized limb minus the change in rating ($T_{+30}$ minus $T_{baseline}$) at the non-sensitized limb. A Mann-Whitney test was used to compare the changes in pinprick sensitivity at the forearm vs. foot.

**Comparison of the area of increased pinprick sensitivity at the forearm and foot.** The radius of the area of increased pinprick sensitivity were compared at both sites using an Independent Samples two-tailed t-test, as the Shapiro-Wilk test did not indicate a significant deviation from normality.

In addition, we assessed pinprick sensitivity at multiple time points up to 4 hours after HFS, with the aim of assessing and comparing the time course (duration) of the HFS-induced secondary hyperalgesia at the volar forearm and the foot.

**Duration of HFS-induced changes at the forearm and foot.** The duration of the HFS-induced change in pinprick sensitivity was estimated as follows. First, at the forearm and foot, each post-HFS measurement of the change in pinprick intensity ratings was (i) expressed as the difference relative to baseline and the contralateral limb as described in the aforementioned section ([$T_{+30}-T_{baseline}$] at the sensitized site minus [$T_{+30}-T_{baseline}$] at the contralateral site) and (ii) normalized such that a value of 1 corresponded to the maximal increase across all time points, and a value of 0 corresponded to the rating reported at the contralateral arm before HFS. The normalized datasets were then fitted to an exponential decay function [$Y = Y_0 * \exp(-K * X)$] where X corresponded to the time relative to the first post-HFS assessment ($T_{+30} = 0$ min), $Y_0$ to the change in pinprick rating at that time point, and K to the rate of decay. This function was chosen after visualization of the overall time courses across testing sites obtained in the present experiment, as well as the time courses of HFS-induced changes reported by Pfau et al. [28]. The analyses were conducted using GraphPad Prism (Version 8.0.1, GraphPad Software, San Diego, California USA), using a Least Squares Regression. Both parameters $Y_0$ and K were constrained to a value greater than 0. To evaluate whether the time courses of the HFS-induced changes differed at the forearm and foot, the Extra sum-of-squares F test was used to compare the fit obtained with the same decay parameter for both datasets with the fit obtained with separate decay parameters fitted to each of the two datasets. The duration of the HFS-induced change in pinprick sensitivity was expressed as the estimated half-life of the effect, corresponding to ln(2)/K expressed in minutes. The same approach was used to estimate and compare the temporal evolution of the radius of the HFS-induced area of secondary hyperalgesia.

## Results

### HFS-induced increase in pinprick sensitivity at the forearm and foot

Group-level ratings and individual pinprick intensity ratings are shown in Figs 2 and 3. HFS delivered to the volar forearm induced a significant increase in pinprick sensitivity at the HFS-sensitized forearm ($\Delta$NRS $T_{+30}$ minus $T_{baseline}$: median = 15.8 [6.7–45.8, IQR]) compared to the contralateral forearm (−1.7 [−3.8–0, IQR]; Wilcoxon signed-rank test: W = 136, p < 0.001) (see also Table 1). The increase in pinprick sensitivity at the HFS-sensitized forearm corresponded to a 2.1-fold increase relative to baseline. Six subjects out of 16 crossed the 50-anchor from non-painful to painful feeling at pinprick evaluation.

Similarly, HFS delivered to the foot dorsum led to a significant increase in pinprick sensitivity at the HFS-sensitized foot ($\Delta$NRS $T_{+30}$ minus $T_{baseline}$: 10 [5.3–21.8, IQR]) compared to the contralateral foot (−0.83 [−3.4–0, IQR]; Wilcoxon signed-rank test: W = 136, p < 0.001). This increase at the HFS-sensitized foot corresponded to a 1.8-fold increase relative to baseline. Two subjects out of 16 crossed the 50-anchor from non-painful to painful domains of sensation.

At $T_{+30}$, the HFS-induced increase in pinprick ratings at the forearm was not significantly different from the HFS-induced increase in ratings at the foot (Mann-Whitney test: U = 154.5, p = 0.327).

### HFS-induced area of secondary hyperalgesia at the forearm and foot

All participants exposed to HFS at the forearm developed an area of increased pinprick sensitivity 30 minutes after HFS. When HFS was applied to the foot, two participants did not develop an area of increased pinprick sensitivity 30 minutes after HFS. In one of these

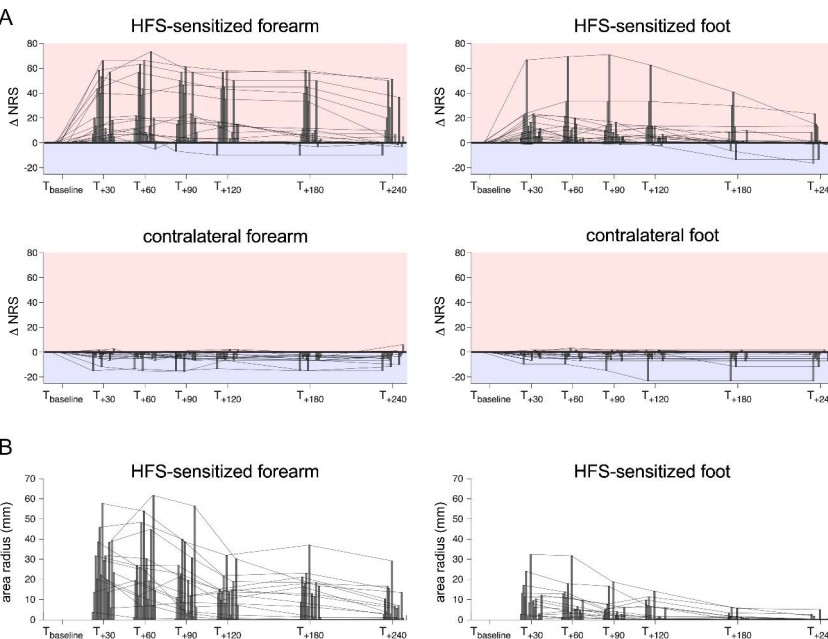

**Fig 2.** A. Change in pinprick sensitivity at the HFS-sensitized and contralateral forearm and foot (individual data). Each vertical bar (and connecting line) corresponds to the difference in pinprick intensity rating relative to the baseline rating ($\Delta$NRS) in one participant. Positive values (light red area) correspond to increases in pinprick sensitivity whereas negative values (light blue area) correspond to decreases in pinprick sensitivity. Note that pinprick intensity ratings increased in almost all participants at the HFS-sensitized forearm and foot, while pinprick ratings tended to remain stable or decreased at the contralateral limb. **B.** Radius of the area of increased sensitivity to pinprick stimulation. Each vertical bar (and connecting line) corresponds to the area radius (mm) in one participant.

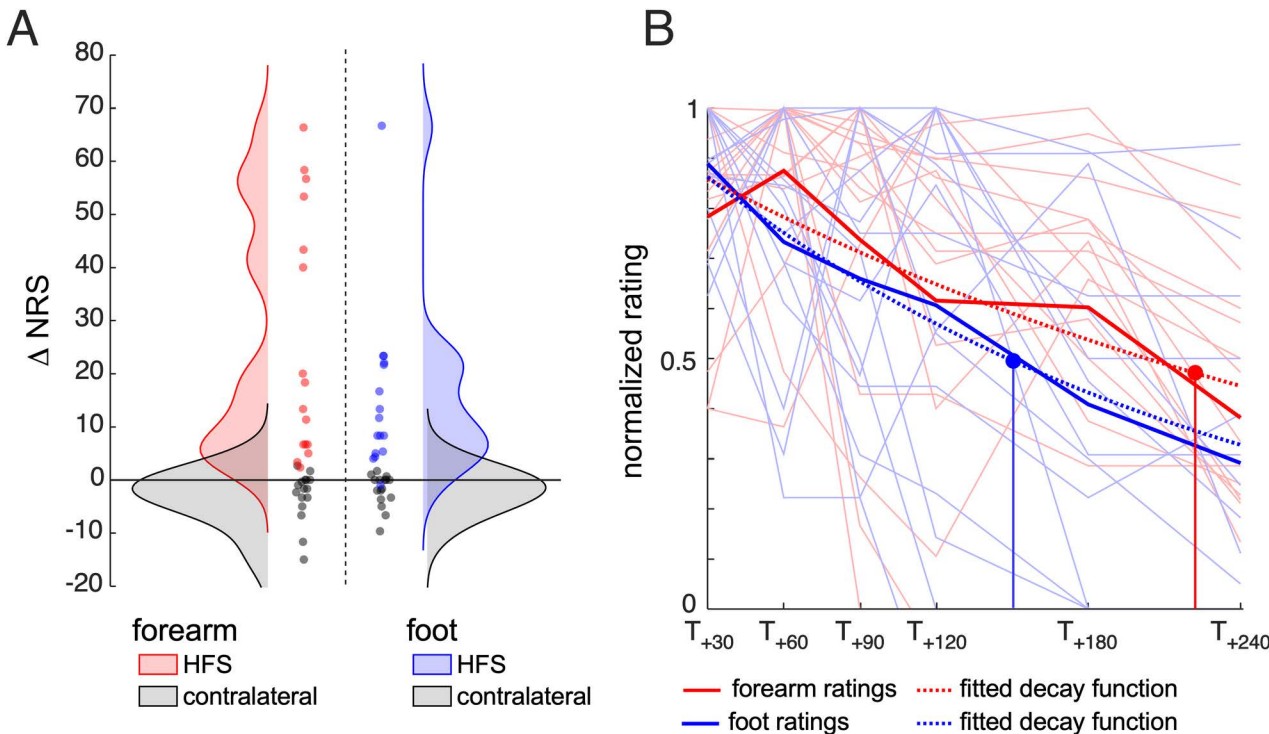

**Fig 3.** A. Density plots of the change in pinprick intensity ratings ($T_{+30}$ minus $T_{baseline}$) at the HFS-sensitized forearm (red), the HFS-sensitized foot (blue) and the corresponding contralateral sites (grey). Individual values are shown as dots. **B.** Normalized individual pinprick intensity ratings (thin light-colored lines) and group-level average (thick lines) at the HFS-sensitized forearm (red) and foot (blue) over time. The fitted exponential decay functions are shown as dashed waveforms. The estimated half-life was 221 min and 150 min at the volar forearm and foot, respectively.

**Table 1. Median and interquartile range of the electrical detection thresholds to a single electrical pulse at the HFS-sensitized forearm and foot. Pinprick ratings (NRS) before HFS (baseline) and 30 minutes after HFS ($T + 30$) at the HFS-sensitized and contralateral forearm and foot, and its estimated half-life. Median and inter-quartile range of the radius of the area of increased pinprick sensitivity 30 minutes after HFS and its estimated half-life, at the HFS-sensitized forearm and foot.**

|  | Forearm | | Foot | |
|---|---|---|---|---|
|  | HFS-sensitized limb | Contralateral limb | HFS-sensitized limb | Contralateral limb |
| **Electrical detection threshold (mA)** | 0.14 [0.06–0.16] | – | 0.32 [0.22–0.35] | – |
| **NRS at baseline** | 16.7 [14.2–22.5] | 15.0 [10.0–18.3] | 10.3 [10.0–13.3] | 10.8 [10.0–13.3] |
| **NRS at $T+_{30}$** | 34.2 [25.8–65.0] | 10.8 [0.2–15.8] | 20.8 [16.3–33.3] | 10.0 [8.2–12.5] |
| **Factor of NRS change** | 2.1 [1.4–3.4] | 0.9 [0.7–1.0] | 1.8 [1.6–2.3] | 0.9 [0.7–1.0] |
| **NRS Half-life (min)** | 221 | – | 150 | – |
| **Radius at $T+_{30}$ (mm)** | 38 [32–46] | – | 22 [12–26] | – |
| **Radius Half-life (min)** | 87 | – | 53 | – |

two participants, an area of increased pinprick sensitivity was reported at the next time point ($T_{+60}$). Group-level and individual radii of the area of increased pinprick sensitivity and their time courses are shown in Figs 2 and 4.

At $T_{+30}$, the mean radius of the area of HFS-induced increased pinprick sensitivity was significantly greater at the volar forearm (median: 38 mm [32–46, IQR]) compared to the foot dorsum (22 mm [12–26, IQR]; Student's t-test: t(30) = 4.2, p < 0.001).

### Time-course of the HFS-induced change in pinprick sensitivity at the forearm and foot

Group-level and individual time courses of the change in pinprick sensitivity and the area of secondary hyperalgesia at the forearm and foot are shown in Figs 2 and 3.

The Extra sum-of-squares F test showed that the decay over time of the HFS-induced increase in pinprick sensitivity was not significantly different at the forearm and foot (F (1, 186) = 1.862, p = 0.174). The decay parameter fitted separately to each dataset was K = 0.003 [0.002–0.005, 95% CI] for HFS delivered to the forearm and K = 0.005 [0.003–0.006, 95% CI] for HFS delivered to the foot, corresponding to a half-life of 221 minutes at the forearm and 150 minutes at the foot.

### Time-course of the HFS-induced area of secondary hyperalgesia

Group-level and individual time courses of the area of increased pinprick sensitivity at the forearm and foot are shown in Figs 2 and 4.

The F test showed that the decay over time of the area radius differed across the two datasets (F(1,182) = 8.593, p = 0.004). The decay parameter fitted separately to each dataset was K = 0.008 [0.006–0.011, 95% CI] for HFS delivered to the forearm and K = 0.013 [0.010–0.016, 95% CI] for HFS delivered to the foot, corresponding to a half-life of 87 minutes at the forearm and 53 minutes at the foot. This suggests a less persistent area of secondary hyperalgesia when HFS was delivered to the foot dorsum compared to the volar forearm.

## Discussion

The magnitude of the HFS-induced amplification of pinprick sensitivity at the dorsal foot and volar forearm were nearly identical, with ratings of pinprick sensitivity increasing by factors of 1.8 and 2.1, respectively. In contrast, the size of the area of HFS-induced secondary hyperalgesia was significantly smaller at the foot compared to the forearm (radius: 22 vs. 38 mm), and decayed more rapidly over time (half-life: 53 vs. 87 minutes).

A previous study compared the secondary hyperalgesia induced by intradermal capsaicin injection at the foot and forearm [29]. As in our study, they reported a more rapid decline of the area of hyperalgesia at the foot compared to the forearm. The size of the area of secondary hyperalgesia tended to be larger on the foot vs. the forearm in the first 15 minutes, but this difference was not significant.

Biophysical properties differing between foot dorsum and forearm skin may influence the efficacy of transcutaneous electrical stimulation [30,31]. Such differences could influence bioimpedance and, thereby, current densities in the skin surrounding each pin of the HFS electrode. These factors, combined with differences in skin innervation and, possibly, central nervous system differences such as the cortical representation of the foot dorsum vs. the forearm, may explain the higher electrical detection thresholds that we observed at the foot vs. the forearm (0.32 vs. 0.14 mA). Importantly, by adjusting HFS intensity to 20 times the detection threshold estimated at each stimulation site, the magnitude of the post-HFS increases in pinprick ratings were similar at both sites. The HFS-induced increase in pinprick sensitivity may thus be proposed as an index of central sensitization in humans that does not critically depend on the tested body site – provided that intensity of HFS is adjusted according to the site-specific electrical detection threshold.

While the magnitude of the increase in pinprick sensitivity was similar at both sites, the spatial extent of the HFS-induced secondary hyperalgesia was significantly smaller and more rapidly decaying at the foot compared to the forearm. Several factors could contribute to this difference, considering the fact that the two stimulation sites did not only differ in terms of

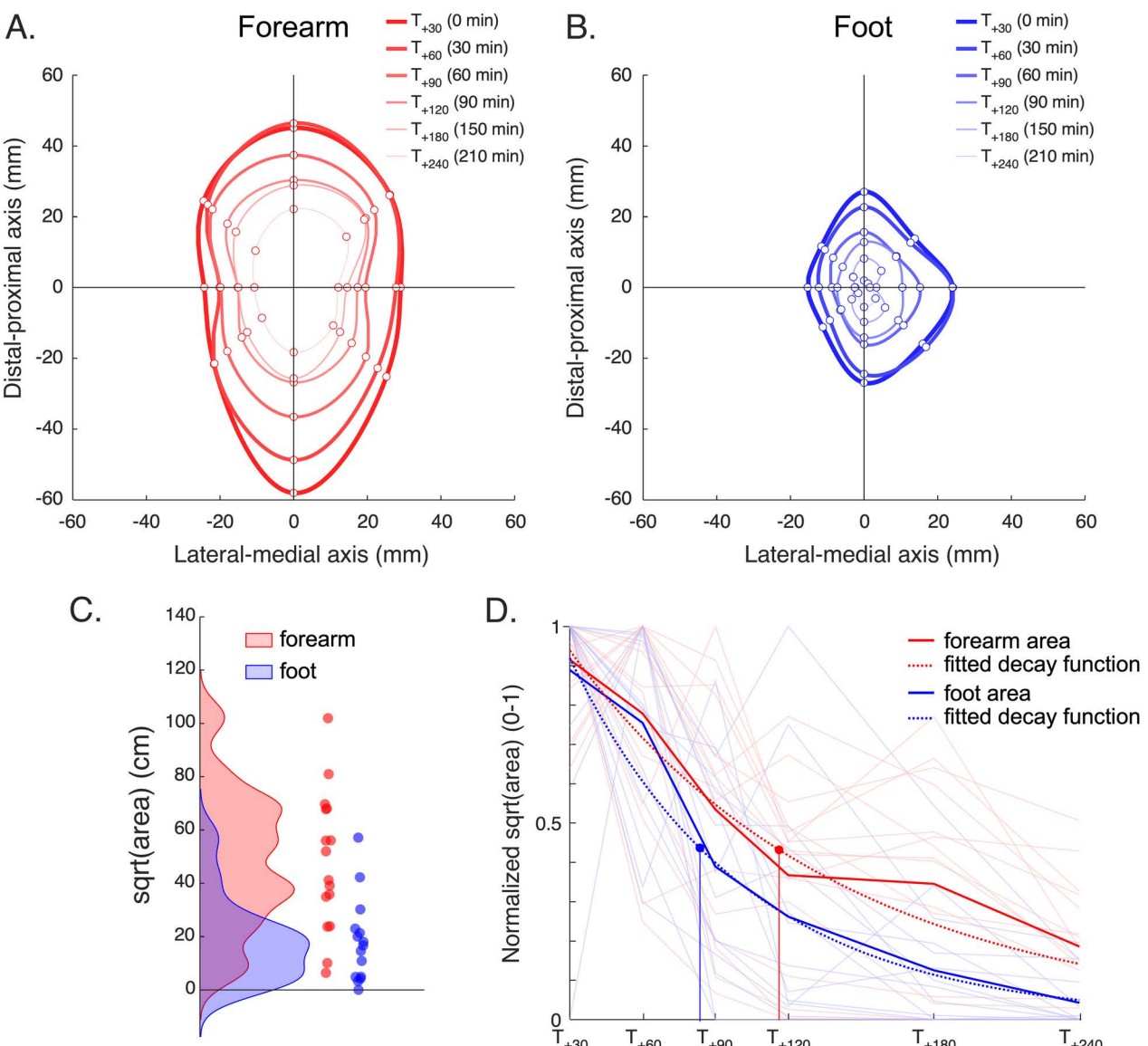

**Fig 4. A.** Group-level average spatial extent of the area of increased pinprick sensitivity assessed at each time point ($T_{+30}$, $T_{+60}$, $T_{+90}$, $T_{+120}$, $T_{+180}$, $T_{+240}$) at the HFS-sensitized volar forearm. Coordinates (X = 0, Y = 0) correspond to the center of the HFS electrodes. **B.** Group-level average spatial extent of the area of increased pinprick sensitivity at the HFS-sensitized foot dorsum. **C.** Density plots of the area radius at the HFS-sensitized forearm (red) and foot (blue), 30 minutes after HFS. Individual values are shown as dots. **D.** Normalized individual (thin light-colored lines) and average (thick lines) radii of the area at the HFS-sensitized forearm (red) and foot (blue) over time. The fitted exponential decay functions are shown as dashed waveforms. The estimated half-life was 87 min and 53 min at the volar forearm and foot, respectively.

stimulated limb (upper vs. lower limb) but also in terms of where the stimulation was applied on each limb (proximal vs. distal end of the limb).

Animal studies have shown that HFS induces both "homosynaptic" and "heterosynaptic" plasticity at the level of the dorsal horn [5,6]. While homosynaptic plasticity refers to enhanced transmission at synapses that were directly activated during HFS, heterosynaptic plasticity refers to changes occurring at neighboring synapses not exposed to HFS. The differences that we observed in the spatial extent of HFS-induced secondary hyperalgesia at the foot dorsum vs. the volar forearm and its decay over time might indicate site-specific differences

in how HFS induces changes at neighboring synapses. Several factors could be expected to produce such differences, including variations in innervation density and peripheral branching of TRPV1-expressing nociceptors that convey the nociceptive signals generating central sensitization in the spinal cord, variations in the central branching of these afferents in the spinal cord [32] and/or variations in the number of activated dermatomes due to dermatome overlap.

Skin biopsy comparisons of intraepidermal nerve fiber density have shown greater innervation densities at the upper vs. the lower limb, but also a strong proximal-to-distal reduction of density in both the upper and lower limbs [33–35]. Microneurography studies reported smaller receptive field sizes and radii of axon reflex flares at the foot compared to the leg, indicating less peripheral branching at distal vs. proximal sites [36]. Therefore, and also considering the fact that Henrich et al. [16] reported a much larger area of HFS-induced secondary hyperalgesia at the forearm ($\pm40\,cm^2$) compared to the hand dorsum ($\pm10\,cm^2$), the foot vs. forearm differences observed in the present study could have been driven mainly by proximal-distal skin innervation differences.

In conclusion, our study shows that HFS can be used to characterize the susceptibility to develop central sensitization at different body sites, which may be useful to profile patients with localized or regional pain. Furthermore, the ability to induce secondary hyperalgesia at the level of the foot is of interest because that site is also testable using the nociceptive RIII reflex, and would allow comparisons with rodent data. While the magnitude of the increase in pinprick sensitivity was similar at both sites, the size of the HFS-induced area of secondary hyperalgesia was larger at the forearm compared to the foot. Therefore, if area size is to be used in clinical studies, reference data is needed for each tested site. For clinical conditions affecting the entire body, the volar forearm could be used as a representative test site similar to the use of the calf as test site for skin biopsies. Further studies are needed to evaluate sensitivity to change of the magnitude and area of HFS-induced secondary hyperalgesia.

## Acknowledgments

The study was approved by the local ethical committee and informed written consent was obtained from every subject before the beginning of the tests. We thank Caterina Nava and Alexandre Stouffs for their support during data collection.

## Author contributions

**Conceptualization:** Louisien Lebrun, Emanuel N. van den Broeke, Bernhard Pelz, André Mouraux.

**Data curation:** Louisien Lebrun.

**Formal analysis:** Louisien Lebrun, André Mouraux.

**Funding acquisition:** André Mouraux.

**Investigation:** Louisien Lebrun.

**Methodology:** Louisien Lebrun, Cédric Lenoir, Emanuel N. van den Broeke, Andreas Schilder, André Mouraux.

**Project administration:** Louisien Lebrun, Ombretta Caspani, André Mouraux.

**Resources:** Bernhard Pelz, André Mouraux.

**Supervision:** André Mouraux.

**Validation:** Emanuel N. van den Broeke, André Mouraux.

**Writing – original draft:** Louisien Lebrun.

**Writing – review & editing:** Louisien Lebrun, Cédric Lenoir, Caterina Leone, Emanuel N. van den Broeke, Andrea Truini, Rolf-Detlef Treede, André Mouraux.

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
