## [Decision Letter · Decision Letter 0]

23 Sep 2024

PONE-D-24-19003Strength, extent and duration of secondary hyperalgesia induced by high-frequency electrical stimulation of the foot compared to the volar forearm of healthy human volunteersPLOS ONE

Dear Dr. Lebrun,

Thank you for submitting your manuscript to PLOS ONE. After careful consideration, we feel that it has merit but does not fully meet PLOS ONE’s publication criteria as it currently stands. Therefore, we invite you to submit a revised version of the manuscript that addresses the points raised during the review process.

The manuscript presents a technically sound piece of scientific research, as confirmed by Reviewer #1, with partial agreement from Reviewer #2, who raises concerns about certain aspects of the data support. While Reviewer #1 found the study well-conducted with rigorous experimental design, Reviewer #2 indicated issues with sample size, recommending it be increased to strengthen the conclusions. Statistical analysis was a point of ambiguity, with Reviewer #1 uncertain of its rigor and Reviewer #2 opting not to comment. Regarding data availability, Reviewer #1 confirmed compliance with PLOS’s data policy, but Reviewer #2 noted that not all underlying data were made available. Both reviewers agreed the manuscript is presented in clear and standard English. Reviewer #1 offered several specific recommendations, such as clarifying the frequency of stimuli in the study, improving the consistency of unit measures, and enhancing clarity in figures by distinguishing meanings of dotted lines. Additionally, Reviewer #1 suggested that some conclusions are overly speculative given the study's design and requested more details on protocol registration and statistical planning. Overall, the manuscript is recognized for its technical merit, but some revisions and data enhancements are advised.

We look forward to receiving your revised manuscript.

Kind regards,

Mohammad Sarif Mohiuddin

Academic Editor

PLOS ONE

Journal Requirements:

1. When submitting your revision, we need you to address these additional requirements.Please ensure that your manuscript meets PLOS ONE's style requirements, including those for file naming. The PLOS ONE style templates can be found at  https://journals.plos.org/plosone/s/file?id=wjVg/PLOSOne_formatting_sample_main_body.pdf and https://journals.plos.org/plosone/s/file?id=ba62/PLOSOne_formatting_sample_title_authors_affiliations.pdf.

b) If there are no restrictions, please upload the minimal anonymized data set necessary to replicate your study findings to a stable, public repository and provide us with the relevant URLs, DOIs, or accession numbers. For a list of recommended repositories, please see https://journals.plos.org/plosone/s/recommended-repositories. You also have the option of uploading the data as Supporting Information files, but we would recommend depositing data directly to a data repository if possible.

Reviewers' comments:

Reviewer's Responses to Questions

**Comments to the Author**

1. Is the manuscript technically sound, and do the data support the conclusions?

Reviewer #1: Yes

Reviewer #2: Partly

2. Has the statistical analysis been performed appropriately and rigorously? 

Reviewer #1: I Don't Know

Reviewer #2: N/A

3. Have the authors made all data underlying the findings in their manuscript fully available?

Reviewer #1: Yes

Reviewer #2: No

4. Is the manuscript presented in an intelligible fashion and written in standard English?

Reviewer #1: Yes

Reviewer #2: Yes

5. Review Comments to the Author

Reviewer #1: Thank you for the opportunity to review this well-written manuscript. The setup seems complex, as do some aspects of the statistical analysis.

Was the protocol of the study published in advance? If so, where can it be accessed? Information on this should be included in the manuscript.

Was there statistical planning to determine the number of subjects needed?

Lines 141-143: What was the frequency of the applied stimuli? You state that you never used the same location twice, however wind-up could also occur if stimuli are applied in a quick repetitive execution within a small area.

eg. Lines 255-257 Reporting of the statistical data should probably always note if IQR or CI is provided.

Lines 305-310 I think this is too speculative ("thus"), your study does not seem to be designed to draw this conclusion.

Figure 1 shows spellcheck highlighting ("HFS-sensitized limb", Tbaseline)

mult. Figures area radius is shown in mm, discussion within the article mentions cm. Same units would look more homogenous.

Fig 3b, Fig 4d should show the different meaning of the dotted line within the image.

Reviewer #2: The sample size was very poor. It should be increased the sample size to conclude this kind of research work. Although the research work is technically sound, and presented in an intelligible fashion.

6. PLOS authors have the option to publish the peer review history of their article (what does this mean? ). If published, this will include your full peer review and any attached files.

**Do you want your identity to be public for this peer review?** For information about this choice, including consent withdrawal, please see our Privacy Policy .

Reviewer #1: No

Reviewer #2: No

---

## [Author Response · Author response to Decision Letter 1]

18 Oct 2024

To the editor,

Please find hereby each reviewer’s comments addressed.

Reviewer #1: Thank you for the opportunity to review this well-written manuscript. The setup seems complex, as do some aspects of the statistical analysis.

We thank the Reviewer for this comment. In the revised manuscript, we have tried to further streamline the presentation of the statistical analyses (page 8 lines 171-172; 190-192, 202-204).

Was the protocol of the study published in advance? If so, where can it be accessed? Information on this should be included in the manuscript.

We thank the Reviewer for this comment. In this exploratory study, planned comparisons were defined before data collection, but the protocol was not published in advance. Only the decision to compare the time courses of the HFS-induced secondary hyperalgesia using an exponential decay function was taken a posteriori, after visualization of overall time courses across testing sites. This has now been clarified in the manuscript (page 5 lines 86-87).

Was there statistical planning to determine the number of subjects needed?

We thank the Reviewer for this comment. This was an exploratory study rather than a confirmatory study. Effect sizes were not known. Our sample size of 32 participants (16 per group) was similar to the size of the samples used in several previous studies assessing differences in HFS-induced secondary mechanical hyperalgesia such as differences related to the frequency of stimulation (15 and 16 participants in van den Broeke et al., J Neurophysiol 2019) or differences related to the pattern of HFS stimulation (15 participants in Gousset et al., J Neurophysiol 2020).

This has now been clarified in the manuscript (page 5 lines 92-97). Effect size estimates of our key outcomes are reported in the manuscript.

Lines 141-143: What was the frequency of the applied stimuli? You state that you never used the same location twice, however wind-up could also occur if stimuli are applied in a quick repetitive execution within a small area.

We thank the Reviewer for this comment. Stimuli were applied every 5 seconds. This information has now been added to the manuscript (page 7, line 150).

Of note, we propose here a table, showing paired t-test proving that repetitive application of pinprick did not significantly change perceived intensity at group level.

Paired Samples T-Test

Measure 1 Measure 2 t df p

HFS side 1st stim - HFS side 2nd stim -1.027 31 0.312

HFS side 1st stim - HFS side 3rd stim -0.145 31 0.886

HFS side 2nd stim - HFS side 3rd stim 0.778 31 0.443

Control side 1st stim - Control side 2nd stim -1.743 31 0.091

Control side 1st stim - Control side 3rd stim -0.679 31 0.502

Control side 2nd stim - Control side 3rd stim 0.790 31 0.435

Note. Student's t-test.

eg. Lines 255-257 Reporting of the statistical data should probably always note if IQR or CI is provided.

We thank the Reviewer for this comment. This is now specified throughout the manuscript (lines 220, 249, 250, 275, 283, 291).

Lines 305-310 I think this is too speculative ("thus"), your study does not seem to be designed to draw this conclusion.

We agree with the Reviewer. The word “thus” was replaced by “might” line 329.

Figure 1 shows spellcheck highlighting ("HFS-sensitized limb", Tbaseline)

We thank the Reviewer for spotting this mistake which was corrected.

mult. Figures area radius is shown in mm, discussion within the article mentions cm. Same units would look more homogenous.

We agree with the Reviewer. Area radius is now reported in mm throughout the manuscript.

Fig 3b, Fig 4d should show the different meaning of the dotted line within the image.

This information has now been added.

Reviewer #2: The sample size was very poor. It should be increased the sample size to conclude this kind of research work. Although the research work is technically sound, and presented in an intelligible fashion.

We thank the Reviewer for this comment. This was an exploratory study rather than a confirmatory study. Effect sizes were not known. Our sample size of 32 participants (16 per group) was similar to the size of the samples used in several previous studies assessing differences in HFS-induced secondary mechanical hyperalgesia such as differences related to the frequency of stimulation (15 and 16 participants in van den Broeke et al., J Neurophysiol 2019) or differences related to the pattern of HFS stimulation (15 participants in Gousset et al., J Neurophysiol 2020).

This has now been clarified in the manuscript (page 5 lines 92-97). Effect size estimates of our key outcomes are reported in the manuscript.

---

## [Decision Letter · Decision Letter 1]

24 Jan 2025

Strength, extent and duration of secondary hyperalgesia induced by high-frequency electrical stimulation of the foot compared to the volar forearm of healthy human volunteers

PONE-D-24-19003R1

Dear Dr. Lebrun,

We’re pleased to inform you that your manuscript has been judged scientifically suitable for publication and will be formally accepted for publication once it meets all outstanding technical requirements.

Kind regards,

Claudia Sommer

Academic Editor

PLOS ONE

Additional Editor Comments (optional):

Reviewers' comments:

Reviewer's Responses to Questions

**Comments to the Author**

1. If the authors have adequately addressed your comments raised in a previous round of review and you feel that this manuscript is now acceptable for publication, you may indicate that here to bypass the “Comments to the Author” section, enter your conflict of interest statement in the “Confidential to Editor” section, and submit your "Accept" recommendation.

Reviewer #1: All comments have been addressed

2. Is the manuscript technically sound, and do the data support the conclusions?

Reviewer #1: Yes

3. Has the statistical analysis been performed appropriately and rigorously? 

Reviewer #1: Yes

4. Have the authors made all data underlying the findings in their manuscript fully available?

Reviewer #1: Yes

5. Is the manuscript presented in an intelligible fashion and written in standard English?

Reviewer #1: Yes

6. Review Comments to the Author

Reviewer #1: I think the only drawback of this publication is the fact, that the protocol was not published in advance. It should be up to the editor to decide, if this aligns with the goals of the journal. My comments were adequately addressed, I have no further comments.

7. PLOS authors have the option to publish the peer review history of their article (what does this mean? ). If published, this will include your full peer review and any attached files.

**Do you want your identity to be public for this peer review?** For information about this choice, including consent withdrawal, please see our Privacy Policy .

Reviewer #1: No

---

## [Editor Report · Acceptance letter]

PONE-D-24-19003R1

PLOS ONE

Dear Dr. Lebrun,

I'm pleased to inform you that your manuscript has been deemed suitable for publication in PLOS ONE. Congratulations! Your manuscript is now being handed over to our production team.

Kind regards,

on behalf of

Prof. Dr. Claudia Sommer

Academic Editor

PLOS ONE
